# Sarcopenia Is Associated with Postoperative Outcome in Patients with Crohn's Disease Undergoing Bowel Resection

**Diogo Carvalho** [1], **Charlene Viana** [2], **Isabel Marques** [2], **Catarina Costa** [3] and **Sandra F. Martins** [1,4,5,*] 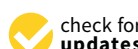

[1] Life and Health Sciences Research Institute (ICVS), School of Medicine, University of Minho, 4700 Braga, Portugal; dvcarvalho_02@hotmail.com

[2] General Surgery Department, Braga Hospital, 4700 Braga, Portugal; vianacharlene@gmail.com (C.V.); isabelmrq@gmail.com (I.M.)

[3] Imagiology Department, Braga Hospital, 4700 Braga, Portugal; anakkosta@hotmail.com

[4] ICVS/3B's—PT Government Associate Laboratory, 4700 Braga/Guimarães, Portugal

[5] Colorectal Unit—Braga Hospital, 4700 Braga, Portugal

[*] Correspondence: sandramartins@med.uminho.pt; Tel.: +351-253-604-828; Fax: +351-253-604-820

**Abstract:** Sarcopenia is as an important prognostic factor in inflammatory bowel disease. In patients with Crohn's disease (CD), sarcopenia has impact on morbidity after surgical resection. Aim: Evaluate sarcopenia impact on prognosis of patients with CD and assess CD sarcopenia prevalence. An retrospective study of 58 CD patients diagnosed histologically and imagiologically at the Hospital de Braga between 1 January 2009 and 31 December 2017. In order to obtain the Skeletal Muscle Index (SMI), it was calculated the muscle area at L3 level, from computed tomography. The *t*-test was used for independent samples, Mann-Whitney test, chi-square test and Fisher's exact test for comparison between groups with and without sarcopenia. Sarcopenia prevalence was 41.4% (24 patients). Patients with sarcopenia presented a muscle area with a mean value of 119.88 $cm^2$ ($\pm$28.10), significantly lower than that of the group of patients without sarcopenia ($t(56) = 2.191$, $p = 0.033$, $d = 0.60$), and values of SMI with median 42.86 $cm^2/m^2$, significantly lower than patients without sarcopenia ($t(56) = 2.815$, $p = 0.007$, $d = 0.08$). Regarding postoperative complications, significant differences were observed between the two groups ($p = 0.000$). In this study, sarcopenia was significantly associated with postoperative morbidity, as reported in the literature.

**Keywords:** Crohn's disease; postoperative morbidity; sarcopenia; skeletal muscle index

## 1. Introduction

Inflammatory bowel diseases (IBD), such as Crohn's disease (CD), are chronic gastrointestinal diseases associated with energy and protein malnutrition [1–4]. Crohn's disease results of a inadequate imunitary response that can involve any part of the intestinal system [5,6]. There has been an increase in CD global incidence, especially in developed countries [7–12]. In Portugal it is estimated that there are approximately 73 patients per 100,000 inhabitants with CD, mostly at the group age of 17–39 [8].

Sarcopenia has been recognized as an important prognostic factor for the morbidity and mortality of patients with IBD [13,14], with a worldwide prevalence of 10% in both sexes [15]. Currently, there is a great deal of interest in testing the correlation between sarcopenia and the impact on the outcome of patients with inflammatory bowel disease.

The first description of sarcopenia dates 1989 by Rosenberg to describe changes in body composition as well as the relationship of muscle mass loss depending on the age [16]. Primary

sarcopenia is related to age, when no other cause is evident, except aging itself, or secondary, when there are causes for it, in addition to aging [17,18]. Sarcopenia is associated with age, activity, disease and nutritional status [17,19]. However, sarcopenia may not fully reflect the general condition and physiological reserves of patients, so there has recently been a growing interest in assessing the influence of body composition parameters on patient outcomes, that is, increased investigation of the clinical importance of secondary sarcopenia [18].

According to the European Working Group on Sarcopenia in Older People (EWGSOP), the definition of sarcopenia consists in the presence of a low muscle mass index and a reduction in muscle function, which is manifested by the decrease in strength and/or physical performance [14,17]. Skeletal muscle volume has a strong correlation with physical performance, measured by gait velocity, as well as muscle strength, clinically characterized by hand grip strength or hip flexion/extension [20–22]. The literature adds that the cross-sectional area of the skeletal muscles at the level of the third lumbar vertebra (L3) is considered a safe marker of total skeletal muscle volume [22]. Thus, the L3 muscle mass index (SMI), that is, the ratio of the cross-sectional area of the skeletal muscles to the level of the third lumbar vertebra (L3) and the squared height, calculated from a CT scan (CT) with axial cut at the L3 level, is used to assess skeletal muscle volume [23–25]. Based on this, a retrospective study was conducted to investigate the relationship between sarcopenia and the prognosis of CD.

With the increase of the average life expectancy, the number of elderly people has been increasing, so as the prevalence of sarcopenia. However, the variability in the definition of sarcopenia in the literature, as well as the cut-off value of the different diagnostic methods, influence this prevalence in the various studies. Yet, according to the EWGSOP, in the age group between 60 and 70 years, the prevalence of sarcopenia is 5–13% and in the group over 80 years, 11–50% [17].

Sarcopenia has been reported as a prognostic factor for outcomes of various diseases, namely CD. Zhang et al., in a multivariate analysis performed in 2015 in patients with CD, demonstrated that the presence of sarcopenia was a significant predicting factor of an outcome with major complications (abdominopelvic abscess, anastomosis dehiscence or peritonitis) after intestinal resection ($p = 0.027$) [26]. It also concluded that, unlike previous studies [14], sarcopenia is not a predictive factor of the need for intestinal resection in patients with CD [26].

The study of sarcopenia is a tool that can easily be introduced in the clinical evaluation of inflammatory bowel disease. This would be a fundamental measure to allow a better understanding of the need for surgical resection [14] and the outcome of patients with CD, regarding postoperative complications. Thus, it became pertinent to evaluate the prevalence of sarcopenia and its impact on morbidity and prognosis in patients with Crohn's disease.

## 2. Results

From 1 January 2009 to 31 December 2017, 58 patients with histological diagnosis of CD performed EnteroTC at the Hospital of Braga. The sample was predominantly female ($n = 31$, 53.4%), with a diagnosis age, between 10.7 and 80.3 years, with a mean of 32.61 ($SD = 14.31$, $Mdn = 29.65$) and a height, ranging from 1.50 to 1.90, with a mean of 1.66 ($SD = 0.08$). According to the Montreal Classification, the disease was localized in terminal ileum in 26 (44.8%) patients, colon in 5 (8.6%) patients, and in 27 (46.6) in ileum and colon. At presentation 49 (84.4%) patients presents with a non-stricturing non-penetrating disease. Regarding the diagnosis, in the majority of cases the it was made from a low digestive endoscopy ($n = 39$, 68.4%), while the other 18 patients (31.5%) CD was diagnosed through EnteroTC.

The sample had a muscular area in L3 with a mean value of 131.85 cm$^2$ ($SD = 36.09$), and a skeletal muscle mass index (SMI) with a median ($Mdn$) of 44.98 cm$^2$/m$^2$ and a mean value of 47.31 cm$^2$/m$^2$ ($SD = 11.63$) (Table 1).

**Table 1.** Demographic data and body composition.

|  | Total *n* = 58 (%) |
|---|---|
| Gender (*n*, %) |  |
| Male | 27 (46.6) |
| Female | 31 (53.4) |
| Age at diagnosis (*M, SD*) | 32.61 (14.31) |
| <16 years | 2 (3.4) |
| <17–40 years | 43 (74.1) |
| >41 years | 13 (22.5) |
| Height (*M, SD*) | 1.66 (0.08) |
| Localization |  |
| Ilion terminal | 26 (44.8) |
| Colonic | 5 (8.6) |
| Ileocolic | 27 (46.6) |
| Upper gastrointestinal | 0 (0) |
| Pattern of disease |  |
| No-nstricturing non-penetrating | 49 (84.4) |
| Fistulating | 5 (8.7) |
| Stricturing | 4 (6.9) |
| Muscular area in L3 (*M, SD*) | 131.85 (36.09) |
| SMI (*M, SD*) | 44.98 (13.42) |

*SD*—standard deviation; *M*—mean; *n*—absolute frequency; SMI—skeletal Muscle mass Index; %—relative frequency.

Regarding laboratory variables, serum albumin presented a mean value of 3.51 g/dL (*SD* = 0.59), and the intervals free of surgery and symptoms showed an average value of 65.74 months (*SD* = 58.84) and 39.95 (*SD* = 63.15), respectively.

Regarding the therapy instituted, it was observed that immunological treatment was used in 77.59% of the cases (*n* = 45), biological treatment in 58.62% of patients (*n* = 34) and corticosteroid therapy in 79.31% of the sample (*n* = 46), and the response to therapy was positive in only 14 patients (24.14%) (Table 2). Concerning the need for surgical resection, it was observed in 24 patients (41.38%), and surgical complications were reported in nine cases, eight anastomosis dehiscence, and one peritonitis.

**Table 2.** Characterization of the sample regarding laboratory, therapeutic, surgical, and postoperative outcome.

|  | Total (*n* = 58) |
|---|---|
| Albumin (*M, SD*) | 3.51 (0.59) |
| Surgery (*n*, %) |  |
| Yes | 24 (41.38) |
| No | 34 (58.62) |
| Surgical complications (*n*, %) |  |
| Yes | 9 (39.13) |
| No | 14 (60.87) |
| Interval free of Surgery (*M, SD*) | 65.74 (58.84) |
| Symptom-free interval (*M, SD*) | 39.95 (63.15) |
| Immunological treatment (*n*, %) |  |
| Yes | 45 (77.6) |
| No | 13 (22.4) |
| Biological treatment (*n*, %) |  |
| Yes | 34 (58.6) |
| No | 24 (41.4) |
| Corticoids (*n*, %) |  |
| Yes | 46 (79.3) |
| No | 12 (20.7) |
| Response to therapy (*n*, %) |  |
| Yes | 14 (24.14) |
| No | 44 (75.86) |

*SD*—standard deviation; *M*—mean; *n*—absolute frequency; %—relative frequency.

From the SMI calculation, the sample was divided into two groups, patients with and without sarcopenia, positive for SMI value less than 38.5 cm$^2$/m$^2$ in women and less than 52.4 cm$^2$/m$^2$ in men. The majority (*n* = 34, 58.6%) had no sarcopenia and its prevalence was 41.4% (*n* = 24). No significant differences were found regarding gender, age at diagnosis and height between groups (Table 3).

**Table 3.** Characterization of sarcopenia in demographic variables.

| | Total (*n* = 58) | Without Sarcopenia (*n* = 34) | With Sarcopenia (*n* = 24) | *p* |
|---|---|---|---|---|
| Gender (*n*, %) | | | | *p* * = 0.425, |
| Male | 27 (46.6) | 14 (41.18) | 13 (54.17) | *Φ* = −0.13 |
| Female | 31 (53.4) | 20 (58.82) | 11 (45.83) | |
| Age at diagnosis (*M, SD*) | 32.61 (14.31) | 32.44 (11.42) | 32.85 (17.90) | *t* (56) = −0.104, *p* = 0.917, *d* = 0.003 |
| Height (*M, SD*) | 1.66 (0.08) | 1.66 (0.09) | 1.67 (0.08) | *t* (56) = −0.611, *p* = 0.544, *d* = 0.02 |

*d*—Cohen's d, effect size measurement; *SD*—standard deviation; *M*—mean; *n*—absolute frequency; *p*—level of significance; *t*—t-test for independent samples; *Φ*—phi, effect size measurement; %—relative frequency; *—*Fisher's exact test.

The individuals with sarcopenia presented a muscular area in L3 with a mean value of 119.88 cm$^2$ (*SD* = 28.10), significantly lower than the group of patients without sarcopenia (*t*(56) = 2.191, *p* = 0.033, *d* = 0.60). Patients with sarcopenia had a SMI value with a median (*Mdn*) of 42.86 cm$^2$/m$^2$, and significant differences were found between the two groups. The mean of SMI in patients with sarcopenia had a value significantly lower than in patients without sarcopenia. (*t*(56) = 2.815, *p* = 0.007, *d* = 0.08) (Table 4).

**Table 4.** Characterization of sarcopenia in muscle quantification.

| | Total (*n* = 58) | Without Sarcopenia (*n* = 34) | With Sarcopenia (*n* = 24) | *p* |
|---|---|---|---|---|
| Muscular area in L3 (*M, SD*) | 131.85 (36.09) | 140.29 (39.01) | 119.88 (28.10) | *t* (56) = 2.191, *p* = 0.033, *d* = 0.60 |
| SMI (*M, SD*) | 44.98 (13.42) | 50.72 (13.00) | 42.49 (7.13) | *t* (56) = 2.815, *p* = 0.007, *d* = 0.08 |

*d*—Cohen's d, effect size measurement; *SD*—standard deviation; *M*—mean; *n*—absolute frequency; *p*—level of significance; *t*—t-test for independent samples; SMI—skeletal muscle mass index; %—relative frequency.

Regarding albumin and symptom-free interval, the group with sarcopenia had a mean serum level of 3.49 g/dL (*SD* = 0.60) and a mean symptom-free interval of 24.30 months (*SD* = 25.49), lower than the group without sarcopenia. No significant differences was found between groups (Table 5).

**Table 5.** Impact of sarcopenia on albumin and symptom-free survival.

| | Total (*n* = 58) | Without Sarcopenia (*n* = 34) | With Sarcopenia (*n* = 24) | *p* |
|---|---|---|---|---|
| Albumin (*M, DP*) | 3.51 (0.59) | 3.52 (0.60) | 3.49 (0.60) | *t* (52) = 0.168. *p* = 0.867, *d* = 0.005 |
| Symptom-free survival (*M, SD*) | 39.95 (63.15) | 47.50 (74.15) | 24.30 (25.49) | *t* (38.4) = 1.150. *p* = 0.139, *d* = 0.04 |

*d*—Cohen's d, effect size measurement; *SD*—standard deviation; *M*—mean; *n*—absolute frequency; *p*—level of significance; *t*—t-test for independent samples.

Regarding the established therapy, it was observed that the immunological treatment was used in 77.59% of the cases (*n* = 45), biological treatment in 58.62% of the patients (*n* = 34) and corticosteroid therapy in 79.31% of the cases (*n* = 46) and the response to therapy was positive in only 24.14% (*n* = 14). No significant differences were observed (Table 6).

**Table 6.** Impact of sarcopenia on instituted therapy and response to therapy.

| | Total (*n* = 58) | Without Sarcopenia (*n* = 34) | With Sarcopenia (*n* = 24) | *p* |
|---|---|---|---|---|
| Immunological treatment (*n*, %) | | | | |
| Yes | 45 (77.6) | 26 (76.5) | 19 (79.2) | *p* * = 0.808, *Φ* = 0.03 |
| No | 13 (22.4) | 8 (23.5) | 5 (20.8) | |
| Biological treatment (*n*, %) | | | | |
| Yes | 34 (58.6) | 19 (55.9) | 15 (62.5) | *p* * = 0.787, *Φ* = 0.07 |
| No | 24 (41.4) | 15 (44.1) | 9 (37.5) | |
| Corticoids (*n*, %) | | | | |
| Yes | 46 (79.3) | 28 (82.4) | 18 (75) | *p* * = 0.527, *Φ* = −0.09 |
| No | 12 (20.7) | 6 (17.6) | 6 (25) | |
| Response to therapy (*n*, %) | | | | |
| Yes | 14 (24.1) | 5 (14.7) | 9 (37.5) | *p* * = 0.064, *Φ* = 0.262 |
| No | 44 (75.9) | 29 (85.3) | 15 (62.5) | |

*n*—absolute frequency; *p*—level of significance; *Φ*—phi, effect size measurement; %—relative frequency; *—*Fisher's* exact test.

Regarding the need for surgical resection, observed in 24 patients, when we divided the groups into patients with and without sarcopenia, it was noticed that there were no significant differences between groups, either for the need of surgical resection or the surgery-free interval. However, the postoperative morbidity were also analyzed in the two groups and significant differences were verified. All patients with sarcopenia who required intestinal resection surgery had some complication, which occurred in only two patients (12.5%) of the group without sarcopenia (Table 7).

**Table 7.** Impact of sarcopenia on the need for surgical resection and postoperative complications.

| | Total (*n* = 58) | Without Sarcopenia (*n* = 34) | With Sarcopenia (*n* = 24) | *p* |
|---|---|---|---|---|
| Surgery (*n*, %) | | | | |
| Yes | 24 (41.4%) | 17 (50%) | 7 (29.2%) | *p* * = 0.176, *Φ* = −0.21 |
| No | 34 (58.6%) | 17 (50%) | 17 (70.8%) | |
| Surgery-free interval (*M*, *SD*) | 65.74 (58.84) | 60.20 (59.38) | 81.43 (59.58) | *t*(21) = −0.753. *p* = 0.460, *d* = 0.04 |
| Morbidity (*n*, %) | | | | |
| Yes | 9 (39.13) | 2 (12.5) | 7 (100) | *p* * = 0.000, *Φ* = 0.825 |
| No | 14 (60,87) | 14 (87,5) | 0 (0) | |

*d*—Cohen's d, effect size measurement; *SD*—standard deviation; *M*—mean; *n*—absolute frequency; *p*—level of significance; *t*—*T* test for independent samples; *Φ*—phi, effect size measurement; %—relative frequency. *—*Fisher's* Exact Test.

Sarcopenia could not be the only factor associated to surgery complications. Therefore, a multivariate analysis was realized, because other factors such as albumin and use of steroids are well known risk factors. The regression model was significant, $\chi2 = 10.31$, $p = 0.016$ (R-squares Nagelkerke = 0.49, percentage of correctly predicted cases = 69.6%). The muscle mass index was found to be a significant predictor of postoperative complications, $p = 0.041$. Lower muscle mass indices are associated with a higher probability of postoperative complications (Table 8).

**Table 8.** Logistic regression model for postoperative complications.

| | OR | *p* | 95.0% CI for OR inferior | superior |
|---|---|---|---|---|
| Albumin | 0.32 | 0.313 | 0.036 | 2.913 |
| Corticoids | 3.53 | 0.495 | 0.094 | 132.543 |
| SMI | 0.83 | 0.041 | 0.691 | 0.992 |

CI—confidence interval; SMI—skeletal muscle mass index; OR—odds ratio.

## 3. Discussion

The factors that promote sarcopenia are multifactorial, including physical inactivity, systemic inflammation, increased metabolic rate and reduced nutrient intake. All these risk factors are prevalent in patients with CD, and the literature on sarcopenia reports values of 60% [19]. In this study, 41.4% of CD patients had sarcopenia. Given the diversity of the methods of analysis and the cut-off value used to define this pathology, the results found in the literature are inconsistent [15]. Nevertheless, studies that opted for the same method of this work, using a cut-off, presents divergent results; two of them revealed a prevalence of sarcopenia of 60% and 61.4% (19.33), while another, also with patients with CD, refers to 37% [14]. Thus, there is a need for further studies to determine the appropriate value for SMI in patients with IBD.

Patients with sarcopenia had a mean muscle area of 119.88 cm$^2$ ($\pm$28.10) and SMI values with a median of 42.86 cm$^2$/m$^2$, value significantly lower than the patients without sarcopenia. Data from the literature support the existence of these differences in both variables [19,26], namely in the SMI, with a median comparable to the one obtained in this study (45.1 cm$^2$/m$^2$) [14]. This finding was expected, since it is from this index that the distinction is made between patients with and without sarcopenia. Regarding gender, age and height, no significant differences were found between the two groups, findings that are supported by literature [19].

When analyzed the differences between patients with and without sarcopenia and post-diagnostic serum albumin, no significant differences were found, on the contrary to what previous articles refer to, where a statistically significant value is described between sarcopenia and serum albumin [14,26]. Since the value of post-diagnostic serum albumin has not been determined in all patients, as in previous studies, and the fact that some patients were already being treated at the date of the first available serum albumin value in the process, may have contributed to the lack of significance. A prospective study, as an alternative to the retrospective study, would be a way to overcome this limitation, since the timing of the collection would be uniform throughout the sample and there would be no patients without albumin values serum.

When we analyzed the impact of sarcopenia on surgery and symptom-free interval no significant differences were observed. However, Bamba et al. concluded in one of his studies that the presence of sarcopenia would have a significant association with the surgery-free period [14]. Because it is a retrospective study, the evaluation of patients' morbidity and symptoms is obtained through clinical records, instead of a directed questionnaire, as would be done in a prospective study, which may have contributed to a bias, and, for lack of significance.

With regard to established therapy, immunological, biological and corticoid treatment, no significant differences were found and the same result was Schneider et al. [19]. Regarding the response to therapy, characterized as the absence of symptoms since the pharmacological institution after the diagnosis, no significant differences were found between the group of patients with sarcopenia and patients without sarcopenia. It was not possible to compare these findings with the literature because there is no study carried out analyzing the impact of sarcopenia on the response to therapy. Bamba et al. exposes the variation of the SMI according to the type of treatment instituted, azathioprine and biological, but does not correlate therapeutics with its symptomatic outcome [14].

Concerning the need for surgical resection, no significant differences were found between the groups of patients with and without sarcopenia. On the other hand, regarding postoperative morbidity, significant differences were found between the two groups. These findings are also corroborated by the literature [25,26]. Zhang et al. found significant differences in the impact of sarcopenia on postoperative complications, contrary to what happens in the need for intestinal resection [26]. For a better analysis of this study, it is important to consider its limitations. Because it was a retrospective study, it was not possible to guarantee a smaller number of missing cases in the statistical analysis of some variables. In addition, a higher sample size might have contributed to the existence of a statistically relevant group analysis. In this case, the use of data from various hospital centers would be a useful approach to overcome this limitation. On the other hand, the use of the first serum albumin

value available after diagnosis, instead of serum albumin coincident with the date of diagnosis, was a bias. Another limitation is that the definition of sarcopenia has been restricted to muscle mass, although muscle function evaluation is also recommended. Nevertheless, studies in the literature that analyzed the influence of sarcopenia in CD used the same approach. Finally, the literature review on this subject is short, which made it difficult to analyze and compare the results.

## 4. Materials and Methods

A retrospective, observational, descriptive and analytical study was carried out. The population covered by the study consisted of patients with histological and imaging diagnosis of Crohn's disease, from 1 January 2009 to 31 December 2017, in Braga Hospital. Ethics approval and consent to participate: This project was approved by Braga Hospital Ethics Committee (CESHB 63/2018; 12 June 2018) and also by Ethics Subcommittee for Life and Health Sciences (SECVS 044/2018; 30 June 2018).

A non-probabilistic sample of convenience was elaborated according to the following inclusion criteria: Patients with histological diagnosis of Crohn's Disease and patients who underwent EnteroCT with complete visibility of the muscle area at L3 level.

Abdominal CT used for diagnosis or the first to be performed post-diagnosis was used to determine the muscle area at the L3 level since at this level there is a good relation with the muscular mass of the whole body, which allows to infer about the total muscular area (TMA) [27]. Using the software ImageJ®(the muscle area was measured at the L3 level, in a single axial section. At this level, the psoas, paraspinal muscles (erector spinae and lumbar quadrate) and muscles of the abdominal wall (transverse abdominal, internal and external oblique and rectus abdominis) are visualized. In order to delimit the muscular tissue, values of Hounsfield Units (HU) of $-29$ to $+150$ were used [27]. The muscle area was delimited manually and calculated automatically by the program.

Sarcopenia was defined to SMI values less than 38.5 cm$^2$/m$^2$ in women and less than 52.4 cm$^2$/m$^2$ in men, according to previous studies by Prado et al. This index is calculated by the following formula [27,28]:

$$\text{SMI} = \frac{\text{Total muscular area} \left(TMA, \text{ cm}^2\right)}{\text{Height}^2 \left(\text{m}^2\right)}$$

*Statistical Analysis*

The quantitative variables were analyzed in relation to the normality of their distribution, based on the asymmetry and kurtosis values, on the Kolmogorov-Smirnov and Shapiro–Wilk test results [29,30]. Not all variables had a normal distribution, so in these cases parametric and non-parametric tests were performed, and once the results were the same, the results of the parametric tests were reported [31].

For the descriptive analysis of the qualitative variables, the absolute *(n)* and relative *(%)* frequencies were calculated. For the quantitative variables, the means *(M)* and standard deviations *(SD)* were presented, and the median *(Mdn)* was also presented when normality was not fulfilled. The comparison between the groups, with and without sarcopenia, was performed through the *t*-test for independent samples *(t)*, and the assumption of homogeneity of the variances was evaluated through the *Levene* test [32]. As a measure of effect size the value of *Cohen's D (d)* was calculated, considering value of 0.20, 0.50, and 0.80 as small, mean and large difference, respectively [33]. For the qualitative variables, the chi-square test ($X^2$) or Fisher's exact test was performed when the percentage of cells in the contingency table that had an expected frequency of less than 5 was higher than 20% [34]. The effect size was calculated using *Phi (Φ)*, since all the variables were dichotomous, assuming the value of 0.10, 0.30, and 0.50 as small, medium and large association, respectively [33]. To analyze the relationship between two quantitative variables, the Pearson correlation coefficient was used.

For the aforementioned tests, statistical significance was considered when $p < 0.05$.

## 5. Conclusions

Implementation of sarcopenia diagnosis is necessary in hospital practice of CD patient's treatment, so standard measures, for prevention and treatment of preoperative sarcopenia, can be performed. This study document a prevalence of 41.4% and an association between sarcopenia and L3 muscle area, SMI, and postoperative morbidity after intestinal resection.

**Author Contributions:** D.C., C.V., I.M., C.C., and S.F.M. designed the structure of the study. D.C. collected the data. D.C. and C.C. performed image analysis. D.C., C.V., I.M., and S.F.M. performed the statistical analysis. D.C. and S.F.M. wrote the final version of the manuscript. All authors read and approved the final manuscript.

**Funding:** The study as no funding support.

**Conflicts of Interest:** The authors declare that they have no conflict of interest. All authors listed have read and approved the submission of the manuscript and have no competing interests.

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
