# Peer review of "Sarcopenia Is Associated with Postoperative Outcome in Patients with Crohn’s Disease Undergoing Bowel Resection"

_gastrointestdisord, doi:10.3390/gidisord1010015_

Round 1

Reviewer 1 Report

Sarcopenia (muscle wasting) is etiologically multifactorial and needs careful explanation to relate it to Crohn’s resection. However, this is an interesting concise and informative manuscript. The manuscript should be in the order of: Abstract, Introduction/Background, Methods, Results, Discussion and Conclusion. Greater attention to syntax and grammar (e.g., making sure sentences are complete) would be appreciated. For all cited studies (results of review discussed in the text), please include the study design, sample size and demographic where relevant, and key findings to enable readers to understand level of evidence of the observation(s).

Please reiterate the purpose of your work, method used, and corresponding results, followed by implications for practice and research (based on previously reported results/discussion points).

In methods please follow standard procedures for conduct and reporting all results of the systematic review. If systematic review procedures were not used, call it what it is. Regardless, search terms, modifier/limits etc. must be reported.

What is needed to increase evidence base for patient follow up recommendations and treatment? Please add a more explicit statement of what clinicians and patients should look for to detect early symptoms of sarcopenia? 

In the discussion, start by summarizing results. The purpose was to explore incidence/rates. What was the result? Was there enough data to know exact rate? The results can be expanded upon and critically reviewed.

Author Response

1.“Sarcopenia (muscle wasting) is etiologically multifactorial and needs careful explanation to relate it to Crohn’s resection. However, this is an interesting concise and informative manuscript. The manuscript should be in the order of: Abstract, Introduction/Background, Methods, Results, Discussion and Conclusion. Greater attention to syntax and grammar (e.g., making sure sentences are complete) would be appreciated. For all cited studies (results of review discussed in the text), please include the study design, sample size and demographic where relevant, and key findings to enable readers to understand level of evidence of the observation(s).”

Answer:  The manuscript order was altered to the suggested one: Abstract, Introduction/Background, Methods, Results, Discussion and Conclusion.

Corrections of syntax and grammar was also realized; and for all cited studies the information requested was introduced when available.

2. Please reiterate the purpose of your work, method used, and corresponding results, followed by implications for practice and research (based on previously reported results/discussion points).

Answer:  The purpose of this work is to evaluate the impact of sarcopenia on morbidity and prognosis in patients with Crohn's disease. Sarcopenia diagnosis.

The diagnosis of sarcopenia will be easily introduced into clinical practice since most of these patients undergo enteroTC during the evaluation of their disease.

3. In methods please follow standard procedures for conduct and reporting all results of the systematic review. If systematic review procedures were not used, call it what it is. Regardless, search terms, modifier/limits etc. must be reported.

Answer:  This information was introduced.

4. What is needed to increase evidence base for patient follow up recommendations and treatment? Please add a more explicit statement of what clinicians and patients should look for to detect early symptoms of sarcopenia?

Answer:  Other studies that prove this association with postoperative morbidity will be necessary, with a larger sample. In addition, studies must be carried out to prove that adequate preoperative nutritional support and physical exercise lead to a better postoperative outcome. Regarding the last question, all preoperative patients should have a screening for sarcopenia.

5. In the discussion, start by summarizing results. The purpose was to explore incidence/rates. What was the result? Was there enough data to know exact rate? The results can be expanded upon and critically reviewed.

Answer:  The alteration was realized.

Reviewer 2 Report

In this study Carvalho et al demonstrated that sarcopenia may lead to severe post-operative complications in patients with Crohn’s disease (CD). Main comments:

1) A linguistic revision is necessary.

2) Line 27: p=0.000 ---> p<0.001

3) Authors should report in table 1 features of CD localization and disease pattern according to Montreal classification.

4) Line 93: which type of immunological treatment?

5) Corticoids --- > corticosteroids.

6) Please give more details about the type of surgery: only resections?

7) Data about disease activity according to Harvey Bradshaw index are lacking. This is a crucial point, since it is presumable that a severe disease may impact on muscle mass, therefore this analysis is necessary.

8) Body mass index (BMI) is another lacking evaluation. Indeed it would be intriguing to explore whether sarcopenia may exist in absence of signs of malnutrition, and if even sarcopenia without malnutrition may impact on surgery outcome.

9) Results in table 4 are obvious, since the diagnosis of sarcopenia was based on muscle area at CT. please delete this table.

10) Table 7: the analysis of surgery free interval should be better performed using Kaplan-Meier curves.

11) Sarcopenia could not be the only factor associated to surgery complications. Therefore it is important to run a multivariate analysis, comprehensive of other factors such as BMI, albumin and use of steroids, which are well known risk factors (see Zhou H et al, Nutr Clin Pract 2018; Galata C et al, Int J Colorectal Dis 2018; Guizzetti L et al, J Crohns Colitis 2018).

12) Line 214: please report the name of the program.

Author Response

In this study Carvalho et al demonstrated that sarcopenia may lead to severe post-operative complications in patients with Crohn’s disease (CD). Main comments:

1)      A linguistic revision is necessary.

Answer:  A linguistic revision was realized.

2)      Line 27: p=0.000 ---> p<0.001

Answer:  Corrected

3)      Authors should report in table 1 features of CD localization and disease pattern according to Montreal classification.

Answer:  As this study is retrospective, this data was not available for a great percentage of patients. However, if it is necessary we will introduce the available data.

4)      Line 93: which type of immunological treatment?   

Answer:  Done

5)      Corticoids --- > corticosteroids.   

Answer:  Done

6)      Please give more details about the type of surgery: only resections?

Answer:  Yes only ressetions was selected.

7)      Data about disease activity according to Harvey Bradshaw index are lacking. This is a crucial point, since it is presumable that a severe disease may impact on muscle mass, therefore this analysis is necessary. 

Answer:  For this study, retrospective, the Harvey Bradshaw index was not possible to introducem as the first 3 questions are answered by patients, in the future with an prospective study it will, for sure, be introduced.

8)      Body mass index (BMI) is another lacking evaluation. Indeed it would be intriguing to explore whether sarcopenia may exist in absence of signs of malnutrition, and if even sarcopenia without malnutrition may impact on surgery outcome.

Answer:  The problem is the same as previous questions, as the study is  reprospective, most patients do not have information about the weight (at the time that enteroTC was realizaed) so BMI cannot be calculated.

9)      Results in table 4 are obvious, since the diagnosis of sarcopenia was based on muscle area at CT. please delete this table.

Answer:  Table 4 was deleted

10)   Table 7: the analysis of surgery free interval should be better performed using Kaplan-Meier curves.

Answer:  Figure 1 with Kaplan-Meier was introduced

11)   Sarcopenia could not be the only factor associated to surgery complications. Therefore it is important to run a multivariate analysis, comprehensive of other factors such as BMI, albumin and use of steroids, which are well known risk factors (see Zhou H et al, Nutr Clin Pract 2018; Galata C et al, Int J Colorectal Dis 2018; Guizzetti L et al, J Crohns Colitis 2018).

Answer:  multivariate analysis wasnot realized because data are few, but if realy necessary we realize it

pede para ler um artigo de Galata de 2018, que associa a albumina pré-operatória como factor de risco de complicações pós-op quando nós não temos o valor de albumina pré-op; IMC não colocado por ausência da variável peso, como explicado em cima.

12)   Line 214: please report the name of the program. 

Answer:  Done

Round 2

Reviewer 1 Report

improved

Author Response

English language and style was check  

Reviewer 2 Report

Answers to point 3-7-8-11 are not satisfactory. Without such data and clarifications, the overall quality of the paper is very low.

Author Response

1) English language and style was checked

2)    table 1 was modified  according to Montreal classification

3)      The Harvey Bradshaw index was impossible to calculate because this is a retrospective study and the  3 questions (of 5 in the total) can be answered by patients 

4)  The BMI can not be calculated because we dont have the infromation of patients weight at the time of the realization of th enteroTC 

5)      As sugested an multivariate analysis was introduced

Round 3

Reviewer 2 Report

Answers are satisfactory